# Hybrid Decision Support to Monitor Atrial Fibrillation for Stroke Prevention

**DOI:** 10.3390/ijerph18020813

**Published:** 2021-01-19

**Authors:** Ningrong Lei, Murtadha Kareem, Seung Ki Moon, Edward J. Ciaccio, U Rajendra Acharya, Oliver Faust

**Affiliations:** 1College of Business, Technology and Engineering, Sheffield Hallam University, Sheffield S1 1WB, UK; nl0716@exchange.shu.ac.uk; 2Materials & Engineering Research Institute, Sheffield Hallam University, Sheffield S1 1WB, UK; b4036163@my.shu.ac.uk; 3School of Mechanical and Aerospace Engineering, Nanyang Technological University, Singapore 639798, Singapore; skmoon@ntu.edu.sg; 4Department of Medicine-Cardiology, Columbia University, New York, NY 10027, USA; ciaccio@columbia.edu; 5Ngee Ann Polytechnic, Singapore 598269, Singapore; aru@np.edu.sg; 6Department of Bioinformatics and Medical Engineering, Asia University, Taichung 41354, Taiwan; 7School of Management and Enterprise, University of Southern Queensland, Toowoomba 4350, Australia

**Keywords:** human and AI collaboration, medical diagnosis support, deep learning, symbiotic analysis process, human controlled machine work

## Abstract

In this paper, we discuss hybrid decision support to monitor atrial fibrillation for stroke prevention. Hybrid decision support takes the form of human experts and machine algorithms working cooperatively on a diagnosis. The link to stroke prevention comes from the fact that patients with Atrial Fibrillation (AF) have a fivefold increased stroke risk. Early diagnosis, which leads to adequate AF treatment, can decrease the stroke risk by 66% and thereby prevent stroke. The monitoring service is based on Heart Rate (HR) measurements. The resulting signals are communicated and stored with Internet of Things (IoT) technology. A Deep Learning (DL) algorithm automatically estimates the AF probability. Based on this technology, we can offer four distinct services to healthcare providers: (1) universal access to patient data; (2) automated AF detection and alarm; (3) physician support; and (4) feedback channels. These four services create an environment where physicians can work symbiotically with machine algorithms to establish and communicate a high quality AF diagnosis.

## 1. Introduction

Cerebrovascular accidents, commonly known as strokes, are the second most deadly disease and a leading cause of disability [1]. Ischemic stroke is the most common type of stroke, which accounts for ≈80% of all strokes [2]. This type of stroke occurs when the bloodstream, to any part of the brain, is blocked by blood clots [3]. When this occurs, brain tissue might get damaged, because the oxygen supply is interrupted. That damage can result in death or disability. Around 75% of all strokes happen in people aged 65 years or older. A meta study from 2009 shows that, within one year, 20,000 U.K. citizens, aged 45 years and below, had a stroke [4]. Worldwide, stroke causes around 5.7 million deaths annually, while in the U.K., around 150,000 people suffer a stroke per year, out of which 53,000 people die [5]. The incidence rate of stroke in males is about 9% of the overall deaths in the U.K., and the same measure for women is around 13% [6]. The Framingham Heart Study showed a connection between Atrial Fibrillation (AF) and ischemic stroke [7]. To be specific, the severity of strokes, in people with AF, is higher, and a stroke has a worse outcome for people with AF when compared to people without AF. AF increases the probability of having a stroke by fivefold, when compared to subjects without AF [7]. The link between AF and stroke is significant, because AF is the most common heart rhythm (arrhythmia) disorder, which affects about 1% of the population [8]. The prevalence of AF increases with age [9,10]. NHS England estimates that only about 79% of all AF cases are diagnosed [11]. One reason for this low detection rate comes from the fact that AF is diagnosed based on heart rhythm irregularities, and these irregularities might be intermittent (paroxysmal) [12], while some forms of AF are even asymptomatic [13]. If an observation coincides with a symptom-free period, then the disease cannot be diagnosed. Hence, a reliable AF diagnosis requires long-term monitoring of the human heart [14,15].

Long-term AF monitoring can be done by measuring the electrical activity of the human heart via a non-invasive Electrocardiogram (ECG). So-called Holter monitors are used for this task, and the resulting ECG measurements are most often used for AF detection [16]. However, the measurement setup is complex because electrical signals are susceptible to noise. Twelve electrodes are routinely deployed by specialized technicians during ECG measurements [17]. Furthermore, ECG signals have a high data rate, which makes them difficult and expensive to distribute and process in real time. Using Heart Rate (HR), instead of ECG signals, can help to overcome these difficulties [18]. As such, HR signals are composed of Beat-to-Beat (RR) intervals. Detecting only the R peak makes the measurement setup less susceptible to noise and hence less complex. Furthermore, a heartbeat occurs about once every second; hence, an HR signal communicates around one sample per second. Compared to the 256 samples a second, used to represent ECG signals, HR signals have a significantly lower data rate. Therefore, HR signals can be communicated easily and inexpensively via mobile networks. There is a large body of literature that establishes that HR signals can be used for AF detection [14,19,20,21,22]. However, the interpretation of the noise-like HR signals is difficult. Even physicians struggle to detect AF through visual inspection of the HR waveform. Furthermore, manual HR interpretation results in inter- and intra-operator variability, which deteriorates the diagnosis quality. Hence, computer-based diagnosis support systems are compulsory for long-term cardiac monitoring [23]. Currently, the most promising approach for manual interpretation of HR signals is to extract diagnostically relevant information, in the form of digital bio-markers, from the waveform. Even with the support of digital bio-markers, physicians can only analyse short HR traces, and the analysis can take longer than the heart takes to produce the trace. That makes real-time assessment impossible in a practical setting.

In this paper, we propose hybrid decision support to monitor atrial fibrillation for stroke prevention. The monitoring service offers universal access to patient HR data, automated AF detection and alarm, physician support and a feedback channel to the patients. The service duration is not restricted. That means our service supports an arbitrarily long observation duration, which might help to detect paroxysmal AF cases. The value proposition for the healthcare providers is twofold. From the medical perspective, a long observation duration has the potential to establish a higher AF detection rate in patients who use the service. Furthermore, the unrestricted observation duration allows a physician to monitor the AF treatment’s efficacy indefinitely. The second value proposition comes from hybrid decision support, which leads to efficiency in terms of both time and cost. The reading physician gets involved only if a Deep Learning (DL) algorithm detected a sequence of AF beats in the HR data; at all other times, human intervention is not required. Hence, the AF detection service reduces the time a physician spends on routine screening tasks. Once AF is detected, the service provides information extraction tools to analyse critical sections of the HR trace effectively. The physician can combine the extracted information with other information sources, such as patient records and personal interaction with the patient, to reach a safe and reliable diagnosis. This diagnosis can be communicated via a feedback channel to the patient. The combination of continuous machine analysis and human oversight creates a cost-effective system for hybrid decision support. Executing the AF detection algorithm for real-time monitoring loads a current Central Process Unit (CPU) core by about 50%. This translates into low processing cost if the algorithm runs on a cloud server. Furthermore, the low data rate implies that the wireless heart rate sensors have a low energy consumption, which keeps both the size and cost down. The value propositions focus on the healthcare provider. The patient benefits from the AF detection service through patient-led signal acquisition, unobtrusive HR measurement and peace of mind through real-time HR monitoring and diagnosis.

To support our value propositions, we structure the remainder of the paper as follows. The next section presents the design steps that led to the prototype implementation. Specific emphasis is placed on the Internet of Things (IoT) and advanced Artificial Intelligence (AI) techniques. The Results Section details the service prototype implementation. The Discussion Section provides a comparison between the proposed service and existing solutions on the market. The Conclusions Section summarizes our method and highlights the major points of the discussion.

## 2. Materials and Methods

We used service design principles to analyse and structure the AF detection problem [24,25]. First, we considered the needs of all stakeholders affected by the proposed service [26]. This understanding shapes the requirements for the AF detection service. The next step is to translate the stakeholders’ requirements to system specification for a successful implementation. The validity of this specification was tested with a prototype implementation, which incorporated hybrid decision support. The following sections provide further details on the individual steps that led to the AF detection service creation.

### 2.1. Need Definition

To establish a need definition, it is necessary to introduce the link between AF detection and stroke prevention in more detail. A stroke occurs when there is a lack of oxygen that causes brain tissue to die suddenly [27]. For ischemic stroke, the lack of oxygen is due to a blockage of the arteries that supply oxygen-rich blood to the brain. In most cases, that blockage is caused by plaque debris in the bloodstream. The heart pumps blood, and indeed the debris, towards the brain tissue through arteries with a decreasing diameter. At some point, the debris will block the artery, and that will prevent oxygen supply to the connected brain tissue. The occurrence of plaque debris is linked to the fluid dynamics of the blood flow, which is governed by the beat-to-beat variability of the human heart. The Framingham Heart Study showed that rhythm irregularities, which change the heartbeat variability, increase the stroke risk [28]. In particular, the study found that a rhythm irregularity (arrhythmia) known as AF increases the stroke risk fivefold.

With that background, the first service design step was to identify the key stakeholders and their needs. We found that there are four key stakeholders in the AF detection service. The sole reason for creating the service is the fact that AF exists in patients. Hence, this group has the primary need when it comes to AF detection for stroke prevention. Healthcare providers aim to address that need by creating an appropriate infrastructure. That infrastructure requires investment based on the cost and the expected benefits. From an abstract point of view, physicians are part of the infrastructure. Their input is crucial when it comes to establishing the benefits of a proposed service. Hence, innovators who create AF detection services for stroke prevention must address the needs of physicians to establish the benefits of their method. However, the effort spent in addressing these needs must be balanced with the required profitability for a practical problem solution. Table 1 details the need definition results.

### 2.2. Requirements Analysis

Based on the need definition, we captured the required functionality and the associated value proposition. Table 2 summarizes both the requirements and value propositions. Cost efficiency and decision support quality are the two most important requirements, because they determine if the proposed service can be used to improve and extend existing infrastructure. All subsequent requirements are functional requirements that answer the question: What service do we build? An alarm message should only be sent when AF is detected. This requirement reflects the information refinement and management nature of the service. An alarm message has a high information content, but a low data rate. This functional specification addresses the requirement for reducing the physician workload. To be specific, the work to establish a suspicion that AF is present has shifted from humans to machines. The AF detection service is a diagnosis support tool, which means all diagnostic decisions lie with the physician. To support that decision, the AF detection service must provide evidence that leads to the suspicion that there is a disease present. This can help to ensure both the functional safety and quality of the diagnosis. It should be possible to provide evidence even if there is no alarm message. This can help during root cause analysis and to improve the service. For example, the proposed service failed to detect AF in a specific patient. Having the ability to retrieve evidence in the form of raw signals might help to establish what caused that fault. That root cause analysis result is the first step to improve the algorithms that provide hybrid decision support. The proposed service should also provide a feedback channel that allows the service provider to communicate with the patient. That channel can be used to disseminate diagnosis results and send to messages that help with patient compliance.

To get a better understanding of the functional requirements of the proposed service, we visualized the service requirements as a sequence of interrelated actions; see Figure 1. These actions were orchestrated along a timeline to create a relatable structure that orders the individual events. The timeline starts with the healthcare provider, represented by a nurse, registering a patient with the AF detection service. Once registered, the patient captures heart rate measurements, which are relayed via a smartphone to a cloud server [29]. In the cloud server, the data are stored and analysed by a DL model [30]. When the analysis results indicate that symptoms of AF were found in the HR data, the cloud logic will send an alarm message to the assigned physician. That message is sent within 5 min of the AF event. In response to the alarm message, the physician will review the evidence contained in the HR trace and fuse this information with further knowledge and experience concerning the patient, in order to reach a diagnosis. If the diagnosis is negative, i.e., the physician decides the patient does not have AF, monitoring for AF continues. Once AF is diagnosed, treatment can be initiated. The treatment efficacy can now be monitored with the same system setup. If AF is diagnosed again, treatment can be adjusted, and the monitoring continues. The next section details the functional specification that was created to meet the system requirements.

### 2.3. Specification Refinement

The specification establishes how the AF detection service is built. This is done by refining the requirements and thereby increasing both the clarity and rigour of the documentation. The AF monitoring is done by detecting disease related changes in HR signals. These signals are easy to measure, cost efficient to communicate, as well as resource efficient to store and process. Hence, this refinement addresses the cost efficiency requirement for the proposed service [31]. Using HR signals provides the foundation for the functional specification. We structured the functional specification into six service components. The following list details how to build these service components:(i)Smart device activation:The smart device activation service enables a patient’s device to activate and establish an account with the healthcare provider. At the start of the service subscription, the healthcare provider registers the patient with the database on a cloud server. The unique account contains patient information. The necessary fields are: patient ID, assigned physician, service start date, service end date. The registration will provide the cloud server login key. This login key is used for both user authentication and data acquisition setup.(ii)Cloud server storage:The patient’s HR data and the DL classification results are stored in the cloud server. This service allows the authorized users to retrieve the data anytime and anywhere.(iii)Real-time HR monitoring service:The patient wears a breast strap with an embedded HR sensor. The sensor picks up the HR signals. These real-time data are displayed on patient smart devices. The patient co-creates value by providing and integrating the data into the AF detection service.(iv)Automated AF detection and alarm service:The DL algorithm analyses patient real-time HR data and classifies the data as AF or non-AF. Once an AF sequence is detected, the system will send an alarm message to the assigned physician. The DL algorithm creates the core value for the system.(v)Physician diagnosis support service:The physician support service incorporates algorithm support in the form of DL results and diagnosis support tools. It helps the physician to verify the DL results and to reach a diagnosis. The value of this diagnosis is twofold. First and foremost, it helps to initiate treatment, which might improve the outcomes for the patient. A secondary use for an established diagnosis arises when we consider improving the DL algorithm. To be specific, a diagnosis becomes the ground truth, which can be used to continuously retrain the DL model. That continued retraining has the potential to improve the detection quality of the algorithm.(vi)Feedback and intervention service:Once the physician has reached a diagnosis, the feedback service can be used to communicate the result to the patient. Social media, email and personal phone calls can be used to provide feedback. Timely appropriate intervention can be carried out to boost the outcomes for patients. Another use for the feedback service is the dissemination of patient compliance messages. For example, through data analytics, it is possible to establish if there is a signal interruption. A compliance message over the feedback channel might help to re-establish the data flow.

## 3. Results

This section describes how we translated the specification into an implementation. The service components were translated into software processes, executed by standard machine architectures and communicated over available infrastructure. Figure 2 visualizes the data flow between different functional entities of the service. The arrangement of the data flow diagram indicates the central role of the cloud storage. The HealthCare app relays the sensor data to the cloud storage. The cluster computing sources the data from the cloud server and, once the data are analysed, puts the result back. The processes are managed based on information from the real-time database. This information is particularly useful to establish the conditions when and to whom an alarm message is sent. This functionality is essential to create the hybrid decision support, which allows medical experts to work efficiently with smart machines. The following sections introduce the functional entities in more detail.

### 3.1. Real-Time Database

The patient information management is based on real-time database entries. During the initial registration process, a representative of the healthcare provider creates a patient record. That record contains patient-specific information, such as the username and password, as well as system-specific information like a cloud server key, which unlocks dedicated data channels. After the initial registration, a patient can use the username and password to login to the HeartCare app. This authentication ensures that the HR measurements are relayed to the patient-specific cloud server channels. The controller node in the cluster uses the patient records to set up the patient monitors, which analyse the HR data in real time. The patient information is also used to manage the alarm message distribution.

### 3.2. HeartCare Mobile App

The AF detection service facilitates patient-led data acquisition. Figure 3 shows a screenshot of the HeartCare app. The graph depicts an averaged HR trace measured with a polar H10 sensor. In that state, the app transfers the HR data to the Thingspeak cloud server [32]. Each patient has a unique API key. Once logged in, the HeartCare app relays the HR data from the sensor to the patient-specific RR_interval_data channel on the cloud server. Both the patient and authorized physicians can access the patient’s data anywhere using the same API key.

### 3.3. Cloud Storage

Each patient account has two cloud storage channels. The first channel, called RR_interval_data, holds the HR measurements. The content is updated when the HeartCare app relays HR signals to the cloud server. The second channel, called AF_detection_ result, holds the DL classification results. The result channel content is updated when the patient monitor produces a new result. Figure 4 shows a patient’s HR data on the Thingspeak cloud server.

Once an AF episode is detected by the DL algorithm, the cloud logic will send an alert to the assigned physician. Sending the alert message can be facilitated with a range of communication channels, such as email, Twitter and instant messages. The message alerts the physician that a dangerous condition has occurred, i.e., AF was detected. The physician decision support and diagnosis service can be used to review the available evidence and to reach a diagnosis.

### 3.4. Patient HR Data Processing in the Cluster

The cluster executes a patient monitor process for each patient. That process network facilitates a real-time data analysis [33]. To accomplish that task, each patient monitor consists of three processes. The first process checks if there is new HR data in the RR_interval_data channel on the cloud. The new data are passed on to the second node, which executes a DL model. The DL results are passed to the third process, which relays them to the AF_detection_result channel on the cloud server.

Processes 1 and 2 of the patient monitor handle the data exchange between the cluster and the cloud server. The main task for the patient monitor and indeed for the AF detection service is real-time HR analysis. We realized this functionality with an Long Short-Term Memory (LSTM) Recurrent Neural Network (RNN) DL model. The model was trained with benchmark data from 20 patients. The data are available from PhysioNet’s [34] Atrial Fibrillation Database (AFDB) [35]. Ten-fold cross-validation established an accuracy of 98.51%, a specificity of 98.67% and a sensitivity of 98.32%, as reported by Faust et al. [14]. A hold-out [36] accuracy of 99% was established with data from three patients. Further hold-out tests established that the DL model could detect AF in unknown HR data with 92% and 94% accuracy for data from LTAFDBand NDSDB, respectively [37]. The physician support module makes the DL results available for physicians in the form of a value ranging from zero to one, which indicates the estimated AF probability. Figure 5 shows the design structure of the proposed DL system. The DL algorithm is composed of three layers, namely bidirectional LSTM, global max pooling and fully connected; for more information about the algorithm, see Faust et al. [14]. The simple structure leaves little space for design errors [38]. Furthermore, the implemented DL algorithm does not require feature engineering. Hence, there is no information reduction due to feature selection, which improves both the accuracy and robustness of the performance results [16].

### 3.5. Physician Support

Physician diagnosis support is a major service component, which was specified in Section 2.3. The implementation of this service component manages the data available on the cloud server. The service component establishes an interface that allows a physician to verify the automated diagnosis results. In other words, the physician can analyse the data and either accept or reject the decision reached by the AI system. We implemented that service component by extending an existing HR analysis and visualization tool. The tool is called the Heart Rate Variability Analysis Software (HRVAS) program, originally developed by Ramshur [39] and published under the GNU public license (https://github.com/jramshur/HRVAS). We extended the program with the ability to download both HR data and the estimated AF probability from the cloud server. Having both, the raw data and the DL results, allows a reading physician to review the available evidence either through visual inspection or through the use of digital biomarkers. For example, visual inspection might reveal fundamental data problems, such as all RR samples having the same value. Digital biomarkers can help to confirm the DL decision result. The ability to establish independent human verification of the machine learning results is a main component for the proposed hybrid decision making process [40].

Figure 6 shows a screenshot of the extended HRVAS program. A drop-down menu allows the user to select the HR signal from a specific patient. The screenshot shows that the signal from Patient 08455 was selected. As such, the signal from that patient was originally downloaded from the AFDB on PhysioNet, and subsequently, it was uploaded to the cloud server [34,35]. The benchmark data allowed us to test the physician diagnosis support service component implementation. The HRVAS Graphical User Interface (GUI) displays the DL results in the upper graph on the left. Displaying the DL results gives an overview of the estimated AF probability, i.e., the reading physician can determine at what time the patient had an increased AF probability. Based on that reading, the physician can select a region of interest and view the HR signal, which corresponds to that region, in the second window. The HR signals trace is coloured in accordance with the estimated AF probability.

Apart from visual signal inspection, the main purpose of the HRVAS program is to visualize digital biomarkers. The workflow unfolds as follows. The physician selects a region of interest on the estimated AF probability graph. Once the region is selected, the corresponding HR trace is displayed, and the digital biomarkers for this region are calculated. The biomarker values are displayed in the right part of the HRVAS GUI. The screenshot in Figure 6 shows time domain biomarkers. The HRVAS documentation provides more details on the available digital biomarkers [39]. These biomarkers are designed to help physicians during the process of validating the DL results and establishing a diagnosis.

### 3.6. Feedback and Intervention

Once the physician has reached a diagnosis, the feedback and intervention service communicates with the concerned patient. Social media, email and personal phone calls can be used to provide feedback. One way to structure the feedback content is a simple traffic light system: green, all is well; orange, take predetermined precautionary action; red, see your physician immediately.

## 4. Discussion

The system reaches a diagnosis through a hybrid decision making process [41]. The hybrid process offers three main advantages: (1) safety through human checks and balances, (2) significantly reduced physician workload, and (3) increased efficiency, which enables real-time diagnosis. The hybrid decision making process is based on analysis results, which are condensed to an independent first opinion on the data [42]. To be specific, we propose a system where an AI algorithm analyses the available data in real time, and a human practitioner only becomes involved if a suspicion is established. However, that design choice is only valid if the AI algorithm is very sensitive when it comes to the detection of AF in HR signals. Another central requirement is cost efficiency. Furthermore, unspecific decision making is not cost effective, because a human expert receives an alarm often, and the machine decisions are routinely overruled. Such unnecessary involvement of human expertise would be inefficient, and indeed, it would be wasteful in terms of time spent rejecting the machine decision, which translates into additional cost for the healthcare provider. Hence, we require the decision support algorithm to have both high Specificity (SPE) and high Sensitivity (SEN). In effect, that leads to a high Accuracy (ACC). Table 3 summarizes research work for the automated detection of AF in ECG and HR signals. The performance measures, reported in the three columns on the right of the table, indicate two points: (1) there is no performance difference between studies based on ECG and HR signals; (2) both the SEN and SPE values are very high. Hence, these algorithms are sufficiently potent to justify large-scale AF detection in a practical service environment.

The proposed AF detection service is based on hybrid decision support, which uses advanced AI for automated AF detection. The high accuracy of this algorithm sets it apart from other solutions currently on the market. The following paragraphs provide some background on current solutions.

An Apple Watch and iPhone combination can be used to detect an irregular pulse. The Apple Watch measures the pulse. Once the signal is captured, an algorithm chain analyses the data. The user receives an alarm message if an irregular pulse is detected. During hold-out validation with benchmark data, that system achieved a positive predictive value of 71% (i.e., only 71% of AF detections by the Apple Watch were actual AF detections; the remaining 28% were not). Based on the same measurements, researchers found that 84% of the participants that received irregular pulse messages had AF. In a subsequent open study, four-hundred thousand users were enrolled. Zero-point-five percent of the participants received irregular pulse messages. Apart from those pulse-based studies, the Apple Watch also features a finger ECG sensor with an AF detection function. However, this only works for as long as the user holds their fingers on the sensor. This may not be long enough to detect AF.

All Apple Watch-based health applications are consumer gadgets, which can establish a suspicion that AF might be present. This suspicion would need to be confirmed by a physician using a heart rate monitoring system.

KardiaMobile with KardiaPro can be used to detect AF at home. The system is based on two electrodes that measure finger ECG. Based on these signals, the device decides if AF is present. In a study with 51 participants, the device had an 8% AF yield, i.e., four people were subsequently diagnosed with AF.

Like the Apple Watch and iPhone combination, KardiaMobile is a gadget that establishes a suspicion that AF is present. For a subscription fee of £58/mo, it is possible to store the ECG data on a cloud service. However, the measurement is not continuous: 30 s ECG snippets are acquired whenever a patient activates the device. Based on such ad hoc measurements, the AF detection algorithm might miss an AF period. If an AF period is detected, the device raises an alarm, and it is up to the patient to interpret that information.

A Holter monitor with software, such as CardioScan, is the gold standard for AF diagnosis and is the standard measurement device used by clinicians. Before a Holter monitor is used, a suspicion is established through the experience of a physician or a gadget. In response to this suspicion, a trained technician will set up the Holter-monitor (place electrodes on the patient’s chest, etc.). Once the setup is completed, the patient wears the device for up to 48 h. The recorded ECG signal is analysed once the device is returned to the issuing clinic. The Holter service costs £50 for a 10 h recording. Apart from the cost, Holter monitors have significant drawbacks. The AF detection rate is positively correlated with the observation interval, i.e., a longer observation interval increases the probability of detecting AF. The data analysis can only start once the Holter monitor is returned; this lack of real-time responsiveness becomes a problem should one choose to increase the observation interval significantly. Wearing a Holter monitor restricts patients’ mobility. If the electrodes detach, the patient must visit the clinic.

Our AF detection service offers long observation intervals and real-time computer-aided diagnosis. The data handling cost is about £30/mo. We envisage that it would replace the Holter system as the clinical gold standard for AF diagnosis. With a positive predictive value of 95.40%, our system achieved a higher AF detection quality when compared to the competitors. The physician support module helps physicians to reach a diagnosis. Establishing a diagnosis and not only a suspicion makes timely intervention possible. Table 4 summarizes the comparison of the AF detection service with three main competitors.

### 4.1. Limitations

In this paper, we outline the design process for a proof of concept AF detection service that incorporates hybrid decision support. As such, this does not yet meet all the stakeholder needs. Before we can offer a complete service monitoring service to patients, the following problems need to be addressed:(i)An alarm message is sent when a dangerous situation arises. Initially, what constitutes a dangerous condition could follow Holter monitoring protocols. For example, an AF event is detected when the estimated AF probability is above 0.5 for at least 30 s [58]. However, it is not known if such an approach is sensitive and indeed specific enough to capture the stroke risk for patients.(ii)Obtaining necessary regulatory approvals (not just the U.K. and EU) especially as regulatory requirements are increasing significantly with the transition to the much more demanding Medical Device Regulations; this can be a long and iterative process.(iii)Negotiating and executing mutually beneficial and sustainable agreements with appropriate commercial partners.(iv)Speed to market; alternative less sophisticated solutions are already available, and new solutions are in development.

### 4.2. Future Work

Addressing the limitations should start with formulating research questions for future work. The proposed hybrid decision support to monitor AF for stroke prevention can help to manage and indeed utilize the real-time information flow that results from extending the observation duration. The prolonged observation duration might lead to new insights about the way in which AF develops in the human body. These new insights should be used to improve and adjust the service functionality. It might be possible to learn and indeed to formulate how human experts interpret the results that lead to a diagnosis. For example, the process generating the alarm message might take into consideration patient age, disease history and severity, as well as the duration of the AF event.

For future work, we propose two clinical studies. The first clinical study is designed to build trust in the technologies that enable the service functionality. We plan to measure HR and ECG from 20 patients at the same time. These measurements will be stored in buffers within the sensors. The ECG analysis results will be considered as the ground truth with which the automated HR analysis results are compared. That will allow us to establish accuracy, sensitivity, and specificity in a practical setting. During the second study, we will focus on fine tuning the clinical processes necessary to deal with real-time HR data. We plan to involve three clinical sites with 20 patients each. We will recruit participants with both known and unknown aetiology to get deeper insights into the link between HR and the nature of embolisms, which might lead to stroke [59]. During that study, a patient is only fitted with one sensor, which communicates HR with a wireless uplink. The wireless uplink will generate a real-time data stream, which is analysed automatically with a DL algorithm. That implies that data are transmitted from the patient environment to a medical cloud server. This will require considerable planning to safeguard the medical infrastructure.

Another aspect for future work is reviewing and potentially influencing the regulatory framework that governs medical decision support systems. Currently, the U.K. (https://assets.publishing.service.gov.uk/government/uploads/system/uploads/attachment_data/file/890025/Software_flow_chart_Ed_1-06_FINAL.pdf) classifies diagnosis support algorithms as medical devices for which certification is required. More work is needed to capture the learning nature of AI algorithms. To be specific, it is not clear how to establish device safety when the functionality changes based on the availability of more data. This is a challenge, not only for the medical device regulation agencies, because retraining the algorithm means changing the decision support model, and hence, the device is not the same as the one that was approved. Initially, a service provider might train new models and have them certified when they show a measurable improvement over the deployed decision support models. In the future, it might be possible to certify the method that retrains the learning algorithm. That would shorten the time for patients to benefit from new decision support models, and it would reduce the administrative effort.

Using the proposed AF detection service for many patients over long time periods leads to big data with reliable labels. With these datasets, it might be possible to gain knowledge about deeper structural properties of AF, such as the relationship with long-term beat patterns and arrhythmias. These structural properties can help to predict and eventually prevent AF for many patients. One prerequisite for this ambitious vision is to create an environment that allows for a continuous retraining of the DL network. Retraining will gradually improve the DL models in terms of detection performance. This will lead to earlier detection of less severe forms of AF. During the retraining process, it might be possible to identify the beat irregularities that indicate AF onset. We might discover something that can be called AF background, because it indicates that the disease is present even when rhythm irregularities are not observed.

The AF detection service’s success depends on the hybrid decision support functionality, which establishes the cooperation among human experts and machines. For the proposed setup, the human expert is firmly in control. Digital biomarkers allow us to establish the validity of the DL result. However, as we move from inference, i.e., detecting AF, to predicting AF, these digital biomarkers and indeed human expertise are less able to carry out that validation task. There might be no human detectable patterns that foreshadow the onset of AF. Hence, the responsibility for the diagnosis shifts towards the machine results. This might be ethically acceptable, because predicting AF implies that we are dealing with a mild form of the disease, which requires only a gentle intervention that causes mild or no side effects. Hence, the role of human oversight might vary depending on the severity of the intervention. For example, a decision to initiate a treatment through anticoagulation should be supported by evidence in the form of physiological signal measurements together with adequate human analysis, because the intervention carries the risk of death. If the intervention consists of a suggestion to change lifestyle choices, such that AF can be avoided, then the requirement for human verification might be minimal. We predict that future hybrid decision support structures will offer such a nuanced validation approach.

## 5. Conclusions

In this paper, we propose a hybrid decision support for stroke prevention based on automated AF detection in HR signals. Commercial HR sensors are used for data acquisition. The sensor data are relayed via a mobile phone to a cloud server for data storage. A DL model evaluates the HR data in real time. The real-time evaluation results take the form of an estimated AF probability. The physician can use that result as a second opinion, which might improve the AF diagnosis, which ultimately leads to a stroke risk stratification. To support physicians during the diagnosis, we incorporate DL results and digital biomarkers in the proposed GUI to provide two independent analysis results. Having two independent results has the advantage that there is no single point of failure, and the digital biomarkers can be used to validate the DL results.

Real-time AF monitoring and diagnosis systems are of great interest because they allow an early diagnosis, which might improve patient quality of life and provide a promising alternative to current healthcare processes. The value propositions focus on the healthcare provider. The patient benefits from the stroke risk monitoring service through patient-led signal acquisition, unobtrusive HR measurement and peace of mind through real-time HR monitoring and diagnosis.

The proposed real-time stroke risk monitoring service has the potential to provide benefits for patients who suffer from heart conditions via accurate automated diagnosis, as well as non-intrusive and uninterrupted treatment monitoring. It also reduces the healthcare cost by replacing expert with machine work. Furthermore, the number of visits to specialized care facilities is kept to a minimum, which benefits the patient and keeps costs low. 

## Figures and Tables

**Figure 1 ijerph-18-00813-f001:**
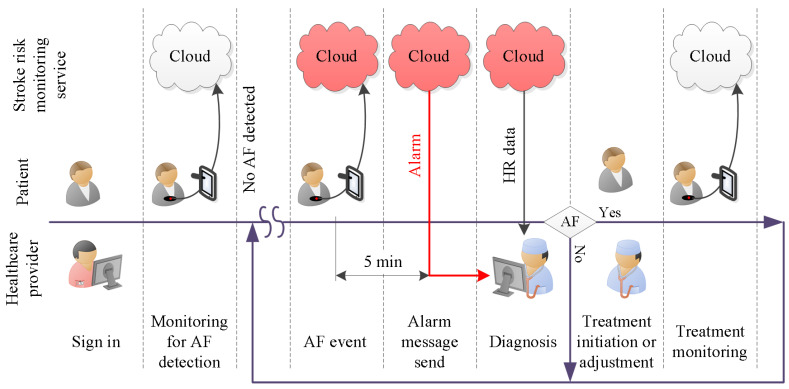
Required service functionality over time.

**Figure 2 ijerph-18-00813-f002:**
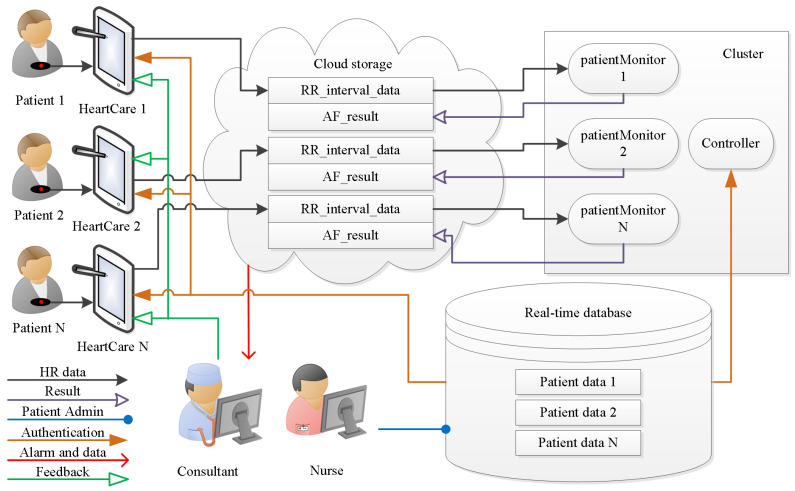
Architecture of the AF detection system for hybrid decision support.

**Figure 3 ijerph-18-00813-f003:**
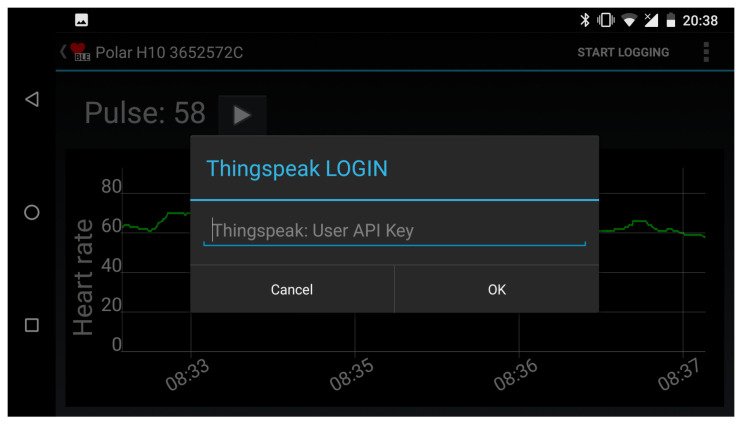
HeartCare app login screenshot.

**Figure 4 ijerph-18-00813-f004:**
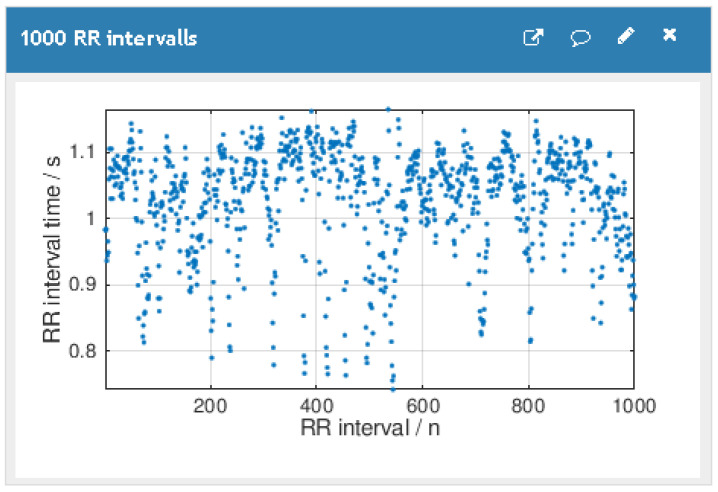
Thingspeak data visualization.

**Figure 5 ijerph-18-00813-f005:**
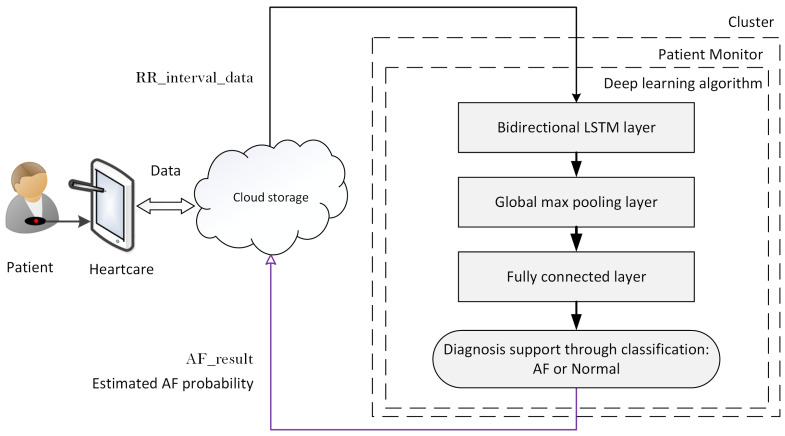
Flowchart of the classification system.

**Figure 6 ijerph-18-00813-f006:**
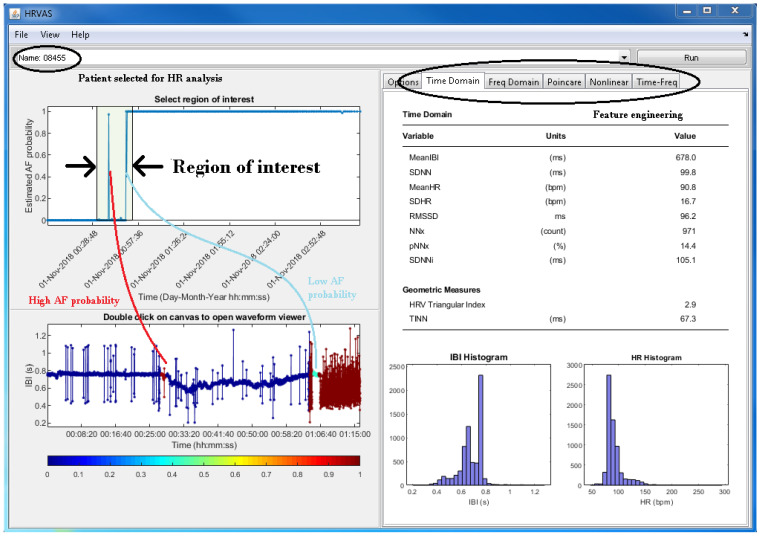
Screenshot of the modified HRVAS program.

**Table 1 ijerph-18-00813-t001:** Stakeholders’ AF detection service with hybrid decision support.

Stakeholders	Needs and Wants
Patients	Reduced stroke risk, less clinical visits, mobility, safety
Physicians	Improved clinical outcomes, high quality diagnosis, safety, reduced workload
Healthcare providers	High efficiency and quality, improved productivity and outcomes, cost effectiveness
Stroke risk monitoring service innovators	Profitability, improved outcome

**Table 2 ijerph-18-00813-t002:** Service requirements and their associated value propositions.

Service	Requirement	Value Proposition
A	Cost efficient and decision support quality	More infrastructure to help a larger number of patients
B	Raise an alarm when AF is detected	Establishing and communicating a suspicion that AF is present in real time
C	Present the evidence for raising the alarm	Providing an overview of the estimated AF probability; this can be used to review the DL results that established a suspicion and triggered an alarm message
D	Allow selecting a time interval of interest; subsequently, the corresponding HR trace can be analysed	Download the HR trace that corresponds to the selected time interval of interest, and calculate features from that HR trace
E	Provide a feedback channel to the patient	Act on the diagnosis by providing appropriate and timely feedback to the patient; act on meta data, such as data stream interruptions, to ensure patient compliance

**Table 3 ijerph-18-00813-t003:** Selected arrhythmia detection studies using HR and ECG. Database (DB) were: MIT-BIH Atrial Fibrillation Database (afdb), MIT-BIH Arrhythmia Database (mitdb), MIT-BIH Malignant Ventricular Arrhythmia Database (vfdb), Creighton University Ventricular Tachyarrhythmia Database (cudb), MIT-BIH Normal Sinus Rhythm Database (nsrdb), MIT-BIH Long Term Database (ltafdb), European ST-T Database (edb) and ecgdb. Hospital data come from non-publicly accessible databases.

Author Year	Method	Data			Performance		
		Type	DB	Rhythm	ACC	SPE	SEN
Faust et al. 2020 [43]	Detrending, ResNet	HR	ecgdb	AF Atrial Flutter (AFL) Normal Sinus Rhythm (NSR)	99.98	100.00	99.94
Ivanovic et al., 2019 [44]	CNN, LSTM	HR	Hospital	NSR, AF AFL	88		87.09
Fujita and Cimr, 2019 [45]	CNN with normalization	ECG	afdb, mitdb, vfdb	AF, AFL, VFDB, NSR	98.45	99.87	99.27
Faust et al., 2018 [14]	LSTM	HR	afdb	AF NSR	98.39	98.32	98.51
Acharya et al., 2017 [46]	CNN with Z-score	ECG	afdb, mitdb, vfdb	AF, AFL, VFIB, NSR	92.50	98.09	93.13
Henzel et al., 2017 [47]	Statistical features with generalized linear model	HR	afdb	AF NSR	93	95	90
Desai et al., 2016 [48]	RQAwith decision tree, random forest, rotation forest	ECG	afdb, mitdb, vfdb	AF, AFL, VFIB, NSR	98.37		
Acharya et al., 2016 [49]	Thirteen nonlinear features with ANOVA with KNN and DT	ECG	afdb, mitdb, vfdb	AF, AFL, VFIB, NSR	97.78	99.76	98.82
Hamed and Owis, 2016 [50]	DWT, PCA and SVM	ECG	afdb	AF, AFL, NSR	98.43	96.89	98.96
Xia et al., 2018 [51]	STFT/SWTwith CNN	ECG	afdb	AF	98.63	98.79	97.87
Petrėnas et al., 2015 [52]	Median filter with threshold	HR	nsrdb, afdb	AF NSR		98.3	97.1
Zhou et al., 2014 [53]	Median filter and Shannon entropy with threshold	HR	ltafdb, afdb, nsrdb	AF NSR	96.05	95.07	96.72
Muthuchudar and Baboo, 2013 [54]	UWT NN	ECG	afdb	AF, VFIB, NSR	96		
Yuanet al., 2016 [55]	Unsupervised autoencoder NN Softmax regression	ECG	afdb, nsrdb, ltdb, hospital	AF	98.18	98.22	98.11
Pudukotai Dinakarrao and Jantsch, 2018 [56]	Daubechies-6 with counters Anomaly detector	ECG	mitdb	AF, VFIB	99.19	98.25	78.70
Salem et al., 2018 [57]	Spectrogram with CNN	ECG	afdb, nsrdb, vfdb and edb	AF, AFL VFIB NSR	97.23		

**Table 4 ijerph-18-00813-t004:** Comparison of the AF detection service with three main competitors.

	Service	Apple Watch and iPhone	KardiaMobile with KardiaPro	Holter Monitor with CardioScan
Performance evaluation				
Quality	PPV: 95.40%	PPV: 71% (pulse)	8% AF yield	N/R
No. of patients	82	N/R	50	N/R
Dataset	AFDB and LTAFDB	Measurement data	Measurement data	Measurement data
System properties				
Signal	Heart rate	ECG	Finger ECG	ECG
Processing	Cloud server	Local	Cloud server	Local
Real-time	Yes	Yes	Yes	No
Diagnosis	Symbiosis between physician and DL	None	None	Feature support
Data storage	Unlimited	None	Snippets	Limited
Model update	Retraining the DL model with cloud data	None	None	None
Use case scenario				
Customer	Healthcare provider	Patient	Patient	Healthcare provider
Physical equipment	Heart rate sensor and Android phone	Apple Watch and iPhone	KardiaMobile device	Holter monitor
Measurement	Patient led	Patient led	Patient led	Expert led
Result	Diagnosis DL decision validated by a physician	Suspicion black box decision; follow-up with Holter recording for diagnosis	Suspicion black box decision; no clear follow-up	Diagnosis established by a physician with analysis support
Limitations				
Diagnosis	HR for diagnosis support is a new paradigm	No diagnosis; diagnosis is established through Holter recordings	No diagnosis	Inter- and intra-observer variability; labour intensive
Safety	Human and machine	Not critical	Not critical	Human
Cost				
Hardware	£ 300	£ 1000	£ 99 and mobile cost	£ 1885.00
Service	£ 30/month	Free	£ 9.99/month	£ 50 for 10 h

## Data Availability

The study did not report any data.

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
