# Peer review of "Hybrid Decision Support to Monitor Atrial Fibrillation for Stroke Prevention"

_ijerph, 2021, doi:10.3390/ijerph18020813_

Round 1

Reviewer 1 Report

This article describes in detail the project of a proof-of-concept prototype for stroke risk monitoring.

The service offers universal access to patient HR data, automated AF detection and alarm, healthcare, and a feedback channel for patients.

The system is based on new technologies: Internet of Things (IoT) technology and a deep learning algorithm that automatically estimates the probability of AF.

AF is one of the leading causes of morbidity and mortality worldwide, and much of this is due to challenges in its timely diagnosis and treatment. Existing and emerging mobile technologies have been used to successfully identify AF in a variety of clinical and community settings, and while these technologies offer great promise to revolutionize AF detection and screening, several major barriers can impede their effectiveness. .

According to the authors' hypothesis, this prototype can offer multiple services: universal access to patient data; AF automatic detection and alarm; Medical attention; and information feedback. Thus, an environment would be created in which clinicians would work symbiotically with machine algorithms to communicate and establish a high-quality diagnosis of AF.

Monitoring the risk of stroke is based solely and exclusively on the detection of FA.

The article brilliantly details the proof of concept, which may be valid, but does not yield any results, so the work can only be considered from an intellectual point of view.

The arguments behind this technology are not new and are already used in other existing medical devices that have been tested and approved by the competent regulatory agencies.

In general, the article is excessively long, especially in the introduction, and could be shortened to facilitate reading.

The introduction provides a good generalized basis for the topic that quickly gives the reader an appreciation for the problem.

The objective is clearly defined in the last sentence of the introduction. However, I think this sentence could be modified, because I think that the prototype does not exactly monitor the risk of stroke, but of AF, they are two very different things. This is the main problem of the article, stroke is a very heterogeneous disease from the etiological point of view, that is, it can be due to many causes and many risk factors are involved, the prototype only measures one. In this sense, it is also convenient to review the title.

The method is appropriate for the study, especially since the main objective of the article is not to develop a stroke risk scale, but to explain a prototype for the detection of FA.

As is, I am sure the article will be considered novel or relevant for publication. However, if the author offers a more detailed discussion, particularly of biological explanations, I think the article could be very interesting and useful for a very wide audience.

Another important issue that needs to be highlighted is the uncertain clinical significance of device-detected AF, potential challenges in integrating patient-created data into existing healthcare systems and clinical workflows, potential resulting harms false positives and identification of the appropriate population range. Investigator-driven screening efforts are potential concerns that warrant a full investigation. The authors did not clearly mention how they will test their prototype, a clinical trial may be necessary.

It is critical that stakeholders, such as healthcare providers, researchers, funding agencies, insurers, and engineers, actively work together to harness the enormous potential of mobile technologies to improve AF identification and management at the facility level. installations. the installations. population. Stroke is a much more complex topic, perhaps the prototype could be tested in a subset of patients with stroke of unknown etiology (Embolic Stroke of Undetermined Source). Authors can review: Stroke. April 2017; 48 (4): 867-872. doi: 10.1161 / STROKEAHA.116.016414.

The six figures and tables show the essential data, but figure 3 appears exactly the same in another publication (check the authorization).

The cited literature is relevant to the study.

Reviewer 2 Report

I am pleased to see advanced data flatform for af detection by accurate realitime based measuring heart rate. However, measuring HR is not only one an indicator to predict for risk of ischemic stroke(too far), hypertension or hypotension blood pressure either. Your flatform Database based realtime HR would be more fit for atrial fibrillation only. So It would be appropriated without stroke monitoring such like table1 etc in a context.

1. Dose it alarm when patients has dradycardia or tachycardia?

Reviewer 3 Report

This manuscript investigated the stroke risk monitoring service for healthcare providers. The following comments can be helpful for the authors to revise and publish the manuscript. 

(1)In the abstract the type of DL algorithem for detection of AF should be explained in detail. As you mentioned in the manuscript, there are lots of published methods for the detection of AF. 

(2) In the abstract, it is not clear how your service is based on the HR measurements. The AF can be detected by the HR only?

(3) What are the advantages and disadvantages of your service compared to the available service in the market?

(4) in table (3), which AF detection method do you think will be useful for your service? 

Reviewer 4 Report

Undoubtedly Atrial Fibrillation is the most common sustained heartbeat disorder in adults. Patients with Atrial Fibrillation have a fivefold increased stroke risk. I agree that early diagnosis, which leads to adequate Atrial Fibrillation treatment, can decrease the stroke risk by 66%.

My comments to the article:

1. The first point called: Introduction is missing from the article.
2. Keyword: "Human and AI collaboration for medical diagnosis" should be divided into several smaller ones.
3. I propose to extend the Introduction by referring to the literature from 2020:
Using Neural Networks for Classification of the Changes in the EEG Signal Based on Facial Expressions, Analysis and Classification of EEG Signals for Brain – Computer Interfaces pp 41-69, 2020 as an introduction to machine learning. This will expand the background of deep learning in the field of biomedical signal analysis. In addition, in the field of IoT, I propose to quote the article: Using BCI in IoT Implementation, Analysis and Classification of EEG Signals for Brain – Computer Interfaces pp 111-128 from 2020, which will update the bibliographies.
4. Figure 5 should be rebuilt so that the graph does not connect to the flowchart. I propose to use Flowchart instead of Flow diagram.
5. The article also requires additional reformatting in terms of tables, so that they fit into the format.
6. Future plans for research as described in the Conclusion section should be expanded.
7. The description in the article shows that: "Submitted to Int. J. Environ. Res. Public Health". However, the article is not reviewed by this MDPI journal.

The authors focused largely on the implementation of development work. The research aspect was developed to a lesser extent.

Round 2

Reviewer 4 Report

Dear Researchers,

The diagram presented in Fig. 2 is not a flowchart, but it is considered legible.

The article still needs to be refined in terms of the final formatting for release.

I accept the explanation that the article was submitted to the special issue "Human and AI Collaborative Decision Making in Healthcare" at the journal International Journal of Environmental Research and Public Health (Int. J. Environ. Res. Public Health).

I recommends the article for publication, after verification of the formatting style, including tables.